# Post-Traumatic Headache in Children after Minor Head Trauma: Incidence, Phenotypes, and Risk Factors

**DOI:** 10.3390/children10030534

**Published:** 2023-03-10

**Authors:** Arianna Dondi, Giovanni Battista Biserni, Sara Scarpini, Anna Fetta, Filomena Moscano, Ilaria Corsini, Greta Borelli, Duccio Maria Cordelli, Marcello Lanari

**Affiliations:** 1Pediatric Emergency Unit, IRCCS Azienda Ospedaliero-Universitaria di Bologna, 40138 Bologna, Italy; 2Specialty School of Pediatrics, Alma Mater Studiorum, University of Bologna, 40126 Bologna, Italy; 3IRCCS Istituto delle Scienze Neurologiche di Bologna, UOC Neuropsichiatria dell’età Pediatrica, 40139 Bologna, Italy; 4Department of Medical and Surgical Sciences (DIMEC), S. Orsola Hospital, University of Bologna, 40126 Bologna, Italy; 5Department of Nephrology, Dialysis and Transplantation, IRCCS Azienda Ospedaliero-Universitaria di Bologna, Alma Mater Studiorum, University of Bolologna, 40126 Bologna, Italy

**Keywords:** post-traumatic headache, minor head trauma, children

## Abstract

Minor head trauma (MHT) is very frequent in children and post-traumatic headache (PTH) is one of its most common complications; however, its management is still a challenge. We aimed to assess the incidence and clinical characteristics of, and risk factors for, PTH among children referred to our pediatric emergency department (PED) for MHT. A total of 193 patients aged 3–14 years evaluated for MTH were enrolled and followed up for 6 months through phone calls and/or visits. PTH occurred in 25/193 patients (13%). PTH prevalence was significantly higher in school-aged (≥6 years) than in pre-school-aged children (21.6% vs. 4.9%, respectively, *p* < 0.009). Females were found to be more affected. The median time of onset was 4.6 days after MHT; resolution occurred in a median of 7 weeks. In 83.3% of patients, PTH subsided in <3 months, while in 16.7% it persisted longer. A total of 25% of children exhibited the migraine and 75% the tension-type variant. Our analysis indicates the presence of headache upon arrival in PED, isolated or associated with nausea and dizziness, as a factor predisposing the patient to the development of PTH. Our findings could be useful to identify children at risk for PTH for specific follow-up, family counseling, and treatment.

## 1. Introduction

Minor head trauma (MHT) is a frequent reason for pediatric emergency department (PED) visits (more than 600,000 visits per year in the USA), accounting for 95% of cases of head trauma (HT) during childhood [1]. It is defined by normal mental status, meaning Glasgow Coma Scale (GCS) scores of 14 to 15 at the initial examination; no abnormal or focal findings on the neurologic examination; and no physical evidence of a skull fracture [2,3]. However, even after a MHT, variable mood, behavioral, and physical changes such as headache, nausea, dizziness, fatigue, hypersomnolence, attentional difficulties, and irritability may appear and persist for a long time, both in children and adults [4,5,6].

About 80% of children with head concussions are reported to have their first visit within the primary care setting, 12% in the PED, and only 5% in a specialty care context [7]. Despite the fact that the majority of these children do not require further testing, pediatric patients with MHT may require a head computed tomography (CT) scan to rule out a clinically important traumatic brain injury (ciTBI). As most children who undergo a head CT scan will have no relevant clinical findings, the Pediatric Emergency Care Applied Research Network (PECARN) scale is an effective tool to identify those patients who actually require this investigation in order to avoid unnecessary radiation exposure [3,8,9,10]. Conversely, children with the presence of isolated loss of consciousness were found to be at very low risk for ciTBI and do not routinely require CT evaluation [11]. In cases of delayed presentation to the PED (>24 h), the same neuroimaging rules could apply [12].

According to the 3rd edition of the International Classification of Headache Disorders (ICHD-3), developed by the International Headache Society, post-traumatic headache (PTH) is a secondary headache that starts within seven days of the injury [13], with a prevalence that varies widely in the literature from 30% to 90% [14,15]. In studies conducted on adults, it is more common as a result of a MHT than of a moderate or severe trauma [16]. Although not always unanimously present among different studies, preexisting conditions such as female sex, familiarity for headache and migraine, history of mental health issues, and insomnia have been considered as risk factors, as well as post-traumatic anxiety and depression symptoms, an immediate headache, and vertigo [17,18,19,20,21,22]. It may present as an acute (if it lasts less than 3 months) or chronic form. The two most common phenotypes of PTH are the migraine and tension-type forms [13]. Cluster headaches and cervicogenic type headaches are also described, but they are rare during childhood also as idiopathic forms [23].

After a MHT, PTH can be one of the most troubling symptoms for both patients and their caregivers [24]. Children with PTH may be challenging and worrisome for the PED physician as well. Many patients have headaches following a HT, and, mainly when symptoms are severe, some of them undergo a head CT scan. However, it is reported that only around 0.5% of these neuroimaging exams have abnormalities related to the HT and the rate of incidental findings unrelated to the trauma is similar to that of the asymptomatic population [25,26]. Despite being one of the most common complaints after traumatic brain injury [27], only a few studies have examined the characteristics of PTH in the pediatric population [28,29] and even fewer have specifically focused on MHT [30], so that the profile of a child with PTH and the precise treatment strategy remain elusive [24].

This study aims to characterize, from a clinical and epidemiological point of view, PTH in a cohort of children with MHT presenting to a PED and to identify any possible predictor for its development.

## 2. Materials and Methods

### 2.1. Recruitment to the Study

For the present study, we prospectively enrolled all patients aged between 3 and 14 years who were evaluated for MHT in our tertiary care PED. Our PED is part of an urban, academic, tertiary care pediatric emergency unit, with approximately 23,000 visits per year of children aged 0–14. Patients older than 14 years are not referred to our PED but to the general ED of our hospital, and were, thus, not included in the study.

Children were enrolled from 10 June 2019 to 1 March 2020; the study was prematurely discontinued due to the onset of the SARS-CoV-2 pandemic.

### 2.2. Inclusion and Exclusion Criteria

Inclusion criteria were age between 3 and 14, and a history of MHT.

Exclusion criteria were age below 3 years because of the difficulty in reporting subjective symptoms; absence of parental consent; language barrier; non-accidental head injury.

### 2.3. Study Design

Data about demographics, patients’ remote medical history, trauma characteristics (mechanism, location), clinical presentation, and management in the PED (triage system priority, GCS, physical examination) were collected into an electronic database. HT was scored according to the PECARN score [8]. Mechanisms of injury were considered high-risk in cases of motor vehicle collision with patient ejection, death of another passenger, or rollover; pedestrians or bicyclists without helmets struck by motorized vehicles; falls > 1.5 m; or head struck by high-impact objects [3]. After discharge from PED, patients were followed up in the following 6 months to establish the incidence and clinical characteristics of PTH. The operational flow-chart is described in Figure 1.

Follow-up assessments were conducted by a research assistant, either on the phone or during a pediatric neurological examination, if necessary or requested by the parents. Data about family history, previous pathologies, timing, pain characteristics, and administered medications were collected. In patients with PTH, parents and patients were required to record a daily headache diary.

PTH was defined as a headache starting within seven days of the injury, and its phenotype was further classified into a migraine or tension-type headache, according to the ICHD-3 [13]. To establish the phenotype, the following features were considered: laterality, pain quality (pulsating or pressing), pain intensity, worsening with physical activity, duration of pain, presence of nausea or vomiting, photophobia and phonophobia, and typical aura symptoms such as photopsia, scintillating scotoma, and hypo/paresthesia. Children complaining of headaches only at the PED immediately after the MHT and not in the following days were not considered to have PTH.

Since the ICHD-3 classification does not mention a frequency criterion for PTH [13], the occurrence of at least one episode per week was considered to define the persistence of headaches beyond the first month.

In the case of a previous personal history of headache, only patients with a worsening pattern of symptoms (any of the following: intensity, frequency of flares, duration, increase in the use of medication) were considered to suffer from PTH.

During telephone interviews with parents, advice on pharmacological and behavioral therapy was provided. In cases of unexplained or worrisome symptoms and signs, or red flags, a pediatric neurological evaluation was promptly planned, or parents were advised to contact their family pediatrician. Red flags included: sudden and violent onset, pain during nighttime or right after awakening, onset or worsening after physical effort, cough, or Valsalva maneuver; chronic progressive pattern of pain; simultaneous presence of fever, projectile vomiting, malaise, unusual behavior, vision changes, seizure, other neurological signs and symptoms; occipital localization of pain [31].

### 2.4. Statistical Analysis

Descriptive statistics of demographic, anamnestic, and clinical variables were used to summarize the data; frequency and percentage for categorical variables, mean and standard deviation for continuous variables. Categorical variables were compared using the χ2, with and without Yates’ correction, when appropriate, and Fisher’s exact test, when appropriate. Relative risk with 95% confidence intervals were also calculated. Patients were considered as eligible for statistical analyses if they completed follow up and listwise correction of other missing data was employed. All statistical tests of a two-sided *p* value of <0.05 were considered significant. Finally, logistic regression with single step approach was used to determine the variables most involved in the development of PTH (dependent variable). The following independent variables were included in the model: age, as the only continuous variable, sex, location of the trauma, mechanism of action, physical findings, associated symptoms at presentation, as binary nominal variables. “IBM SPSS Statistics” for Windows, 22nd version (IBM Corp., Armonk, NY, USA) was used for the analyses.

### 2.5. Ethical Issues

The present study is part of a broader prospective study evaluating the management of MHT in the PED of Scientific Institute for Research and Healthcare (IRCCS) Sant’Orsola University Hospital in Bologna, Italy; the protocol (number 039/2017/O/Oss) was approved by the Local Ethics Committee on 14 February 2017. Informed consent was obtained by the parents or caregivers of the included patients.

## 3. Results

### 3.1. Population and Characteristics of MHT at PED Evaluation

A total of 228 children aged 3–14 years were referred to the PED for MHT, and 224 were enrolled. The number of preschool-age children (younger than six years old) was 99 (65 males; 34 females), whereas the number of school-age children was 129 (87 males; 42 females). Eleven (5%) patients suffered from comorbidities not related to PTH. Table 1 reports population and MHT characteristics.

Thirty-one patients (13.8%) were lost at follow-up either because the phone number was wrong, or did not answer the phone, or refused to answer the questions, and, therefore, were not included in the following statistical analyses that were performed on 193 children. The flow chart of the study population is reported in Figure 2.

### 3.2. Incidence, Timing, and Characteristics of PTH

PTH occurred in 25 of 193 patients (13%). If assuming that all patients lost to follow-up developed PTH, the incidence of PTH would increase to 26.3% (60/228); if, instead, assuming that all patients who were lost at follow-up did not develop PTH, the incidence of PTH would decrease to 10.9% (25/228). One of the patients who developed PTH was lost later at follow-up (Figure 2). Five patients required a pediatric neurological evaluation. Two patients with PTH underwent a head CT scan at the PED assessment, but none required further neuroimaging at follow-up. Clinical characteristics of PTH patients are summarized in Table 2.

The mean time of onset of PTH from MHT was 4.6 days after the event (SD 2.6 days), with two incidence peaks at 1 and 7 days (Figure 3).

The median resolution time of the headache was 7 weeks, with a peak at 8 weeks from the beginning of the condition. A total of 20 cases (83.3%, 10 females and 10 males) showed an acute PTH, lasting less than 3 months; 4 patients (16.7%, 2 females and 2 males) showed a persistent form, lasting up to 6 months after the trauma. PTH persistence over time is reported in Figure 4.

A total of 6 out of 24 (25%) children exhibited the migraine variant (first group), while 18 (75%) exhibited the tension-type variant (second group). Among patients who developed PTH, 21% and 32% had a personal and a family history of headache, respectively.

### 3.3. Risk Factors for PTH

To identify possible risk factors for PTH, patients were divided into two groups according to whether PTH occurred, and cross-tabs were created. Table 3 summarizes the differences between the two subpopulations.

The characteristics reported in Table 3 did not differ in the migraine vs. the tension-type PTH group. Different mechanisms of injury were not significant predictors of PTH development. Analgesic drugs were used more frequently in the chronic than in the acute PTH group (100% vs. 60% of the patients, respectively); however, this difference was not statistically significant. Physical exercise was reported by 21% of the sample as a factor triggering the onset of a PTH episode. The logistic regression based on the possible risk factors for PTH reported in Table 4 shows these can only account for about 30–40% of the variability in PTH incidence.

The model itself, although not significant, shows a classification capacity of 90.2%, gaining about 5% in capacity in predicting PTH incidence. Table 5 shows the risk factors and their individual probabilities of predicting PTH incidence.

## 4. Discussion

To the best of our knowledge, this study is one of the few prospectively focusing on PTH in a pediatric population from both a clinical and an epidemiological point of view. Moreover, while most studies considered children aged 5 years and older [32], we enrolled children over 3 years, managing to provide relevant data on this age group as well.

The selection of a large sample of children presenting for evaluation for MHT to our PED allowed us to provide a more accurate estimate of the prevalence of PTH, setting it at 13%. This result is higher than previous studies in the literature. Moscato et al. [33] studied the PTH incidence for one year in four Italian headache centers: in a population of 1656 patients aged 4–18 years referred for headache, they reported an incidence of 3.2% at first consultation. Another large study of about 670 children referred to the ED for mild traumatic brain injury reports a PTH incidence of 11% 2 weeks after the event [17]. The difference in the overall incidence between studies can be explained by the different features of patients referred to specialty headache clinics and children arriving at the PED for an MHT. It is likely that patients referred to a specialist have already been assessed by their general practitioner or pediatrician, who might have already excluded a PTH diagnosis. In addition, only about 12% of patients with an MHT are referred to the PED, and usually only when the trauma is perceived as severe; thus, we might have selected a population at higher risk than all the children that experience MHT.

Moreover, contrarily to other studies [34], also children with a previous personal history of headache and a worsening pattern of symptoms were considered to suffer from PTH, in consideration of the overlapping pathogenesis between primary headaches (both tensive and migraine types) and secondary ones. Therefore, we considered trauma as a trigger in post-traumatic onset forms and as an aggravator in forms with previous onset [23,35].

In our sample, PTH prevalence was significantly higher in school-aged (≥6 years) than in pre-school-aged (<6 years) children. This observation is consistent with data from several studies showing a greater risk for adolescents to develop symptoms after MHT than pre-adolescent children [33,36,37]. In contrast, in a six-year study based on the first outpatient visits of 1598 children with recurrent and chronic headaches, the incidence of PTH was 17% and 8% in those aged 3–6 and over 6 years, respectively [38]. This difference could be explained both by the different type of enrollment (in our case, children with MHT, in theirs, children with chronic headache), and by the fact that children with PTH usually consult the general practitioner or the family pediatrician, rather than a headache specialized center. It is necessary to underline that the diagnosis of PTH in the lower age groups can be challenging because, for these children, it is harder to report their symptoms, so that the real incidence of PTH might be underestimated. However, this confounding factor is not modifiable, and it may affect all the studies about headaches in the pediatric age. Moreover, older children sometimes have the tendency to overestimate the severity of their symptoms compared to the parents’ description [36].

Females were found to be more affected than males, in accordance with previous observations both on the pediatric and adult patients [33,39,40]. In addition, when considering differences by sex in the different age groups, we found that the incidence of PTH in school-aged females is greater than in school-aged males, preschool males, and preschool females. These observations suggest that being a school-aged female may represent a risk factor for the development of PTH. In the subgroup with chronic PTH, two patients were female.

A positive personal or familiar history of headache was found in patients who developed PTH. This data is consistent with Kuczynski et al. [17], who conducted a prospective longitudinal cohort study of symptoms following MHT in a group of 670 children (aged 0–18 years) presenting at PED; they discovered that a family or past medical history of migraine was present in 82% of children with persistent PTH. The same study identified familiarity for headache and a history of headache preceding the HT as risk factors for developing PTH, in agreement with other works in the literature [17,39,41].

PTH appeared in all children within the first week after HT and, in most cases, resolved within 3 months. In 4/24 (16.7%) of our sample, headaches lasted for at least 6 months. While studies of the adult population report that 58% of patients after traumatic brain injury continue to have headaches one year later [9], the rate in children remains unclear, with few studies reporting a prevalence between 7.8% and 12% at 1 year [17,30,42]. The short observation time of our study does not allow making comparisons with these papers; it is possible that similar percentages would be present after one year.

Our analysis indicates the presence of headache upon arrival at the PED, isolated or associated with amnesia, loss of consciousness, nausea, and dizziness, as a factor predisposing the patient to the development of PTH. In other trials enrolling children, the presence of these symptoms in the PED was strongly associated with the development of PTH [16,43,44,45,46].

This finding could allow better identification of patients at risk for PTH and possibly set up non-pharmacological prophylactic interventions such as nutraceuticals such as magnesium, which was found to be more effective than placebo in a two-arm randomized cohort study in adolescent patients experiencing PTH [47,48].

The incidence of dizziness in our study was overall low (5.8% in all patients undergoing MHT and 8% in those with PTH), in line with other pediatric reports that set it around 7% for patients who develop PTH [18]. Other authors document children complaining of dizziness as being around 20% of those with post-concussion syndrome [36].

Conversely, transient alteration of consciousness or memory loss without headache were not risk factors for PTH, either individually or in combination. A possible association is still debated in the literature [23], even if some authors consider it a risk factor for persistent PTH [34].

No individual factors usually related to the severity of the trauma or symptoms at presentation seem to predict PTH incidence. Consistent with the published literature [39], high-risk mechanisms of the HT were not found to be a significant factor in the onset of PTH in our population, as were head lesions (hematoma, scalp edema, and induration), nausea or vomiting, and gait instability. Therefore, the risk factors included in the PECARN score (except for headache) are inversely related to PTH incidence [8], meaning that the probability of having a ciTBI does not relate to the development of PTH and involves different mechanisms.

The pathophysiology underpinning PTH remains largely unknown, but several possible disease mechanisms overlapping idiopathic headaches have been proposed, including impaired descending modulation, neurometabolic changes, and activation of the trigeminal sensory system [23]. Moreover, there are multiple precipitant factors for headache, and many of these can be experienced by individuals after a concussion or traumatic brain injury (e.g., adverse psychological reactions to the injury-related circumstances, depressed or anxious mood, parental trauma-related anxiety [49], sleep deprivation, interrupted sleep, altered sleep–wake cycles, etc.) [50]. Therefore, the focus of the diagnosis and management of PTH should be on these aspects.

The use of ICHD-3 criteria [13] for the diagnosis of PTH and its phenotypic variants allowed us to classify our populations in a standardized manner: 25% resulted as migraine-type and 75% as tension-type. This result differs from what is reported in the literature, as most studies state that the migraine phenotype prevails over the tension type in PTH, with variable frequencies [17,28,34]. These discrepancies could be due to methodological differences, such as inclusion criteria, PTH classification (which has undergone several changes over the years), and time of evaluation of the patients. We did not find any specific predictive factors for either of the two forms, probably because of the small sample size.

Among the factors triggering the onset of a PTH episode, physical activity was reported by 21% of the sample. In the literature, the return to school and sports in children with PTH has long been debated. A Canadian study compared the outcome at two weeks between a group of children who returned to physical activity 72 h post-concussion, even if symptomatic, and one who returned to physical activity once asymptomatic. Among adherent participants, early physical activity was associated with reduced symptoms at 2 weeks [51]. From our data and those in the literature, therefore, it seems appropriate to recommend rest for the first 2–3 days after a MHT and then gradually return to the normal previous levels of physical activity. Moreover, a recent review proposed an algorithm for a proper, gradual return to normal activity levels, favoring 24 to 48 h of both physical and cognitive rest followed by a return, first, to learning activities (i.e., back to school) and to sports only when a full return to school allows symptom stability (no new symptoms emerge and previous symptoms remain stable) [52].

Our study has some limitations. First, by including only children whose MHT was assessed at the PED, we cannot determine the prevalence and characteristics of PTH in the entire population that suffers from HT. Our study does not include a control group, and the observation time is relatively short. Our sample included twice as many males as females: this reflects the expected variances of MHT in childhood but may also partly bias the reported inflated risk of PTH in females. Regarding PTH characterization, information during follow-up was mostly collected via self-report, whereas physical examination was completed only in a small part of the sample. Moreover, symptoms of preschoolers were interpreted and reported by the parents in most cases. Finally, we did not evaluate differences in drug therapy and treatment outcome, nor did we investigate the occurrence of repeated head injuries during follow-up phone calls.

## 5. Conclusions

In this study, we characterized the incidence, clinical features, and risk factors of PTH in a group of 193 children over three years of age who were referred to our PED because of MHT. The incidence of PTH was 13% and was higher in the school-aged group and in females. Presenting to the PED with a headache as a major complaint was found to be a predisposing factor for the onset of PTH, both as an isolated symptom and in association with other symptoms, especially nausea or vomiting, amnesia, and feeling unstable or dizzy. On the other hand, the presence of a transient change in consciousness, the severity of the HT mechanism, the localization of the trauma in the skull, and the execution of a head CT scan in PED were not significant risk factors. These findings could be useful to identify, among the children referred to the PED for MHT, a group more prone to developing PTH, to formulate a counselling plan for parents, and to specifically follow them. Moreover, advising the parents and caregivers that children should avoid intense physical exercise in the first 1–2 days immediately following the trauma, with a subsequent gradual return to the usual activity levels, might limit the onset of PTH or its duration over time. New studies on larger populations are needed to confirm our results and identify any other factors that can predict PTH onset and phenotype.

## Figures and Tables

**Figure 1 children-10-00534-f001:**
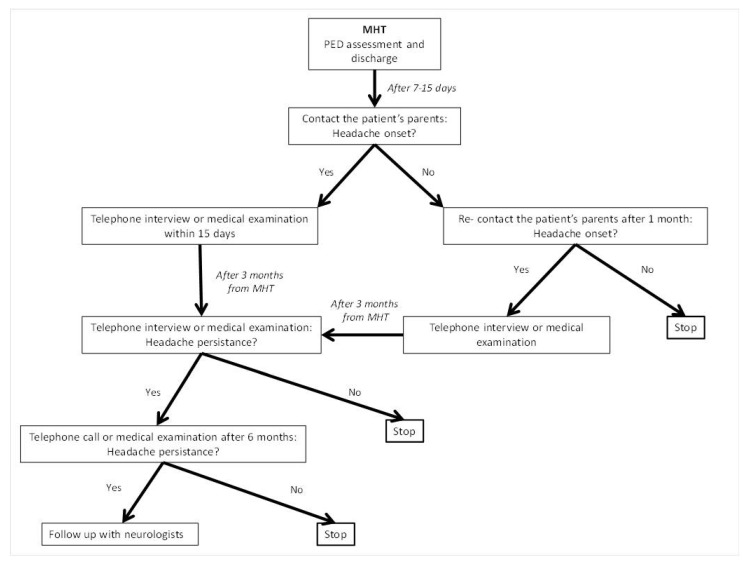
Operational flow-chart of the study.

**Figure 2 children-10-00534-f002:**
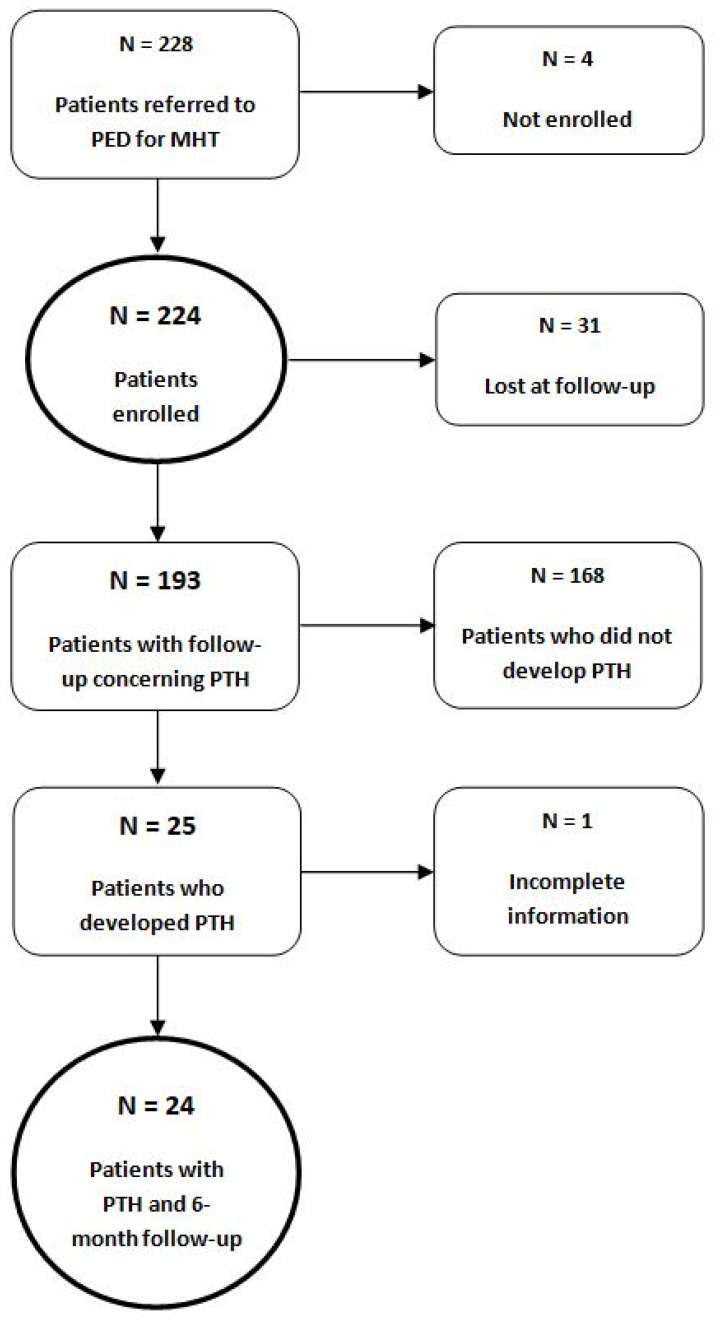
Flow chart of the study population. N = number of patients.

**Figure 3 children-10-00534-f003:**
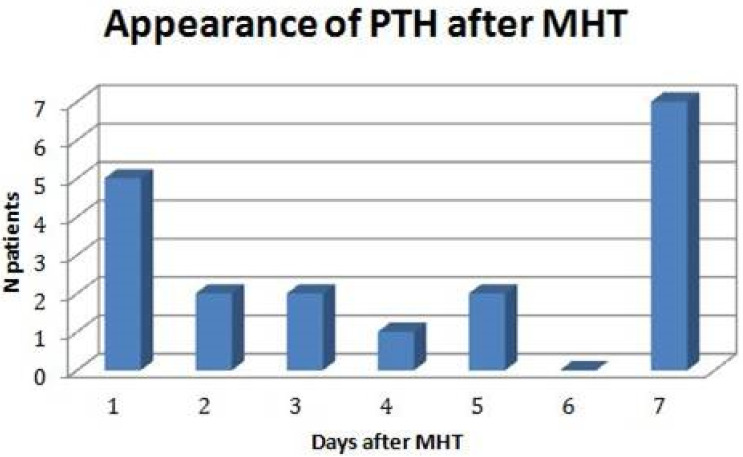
Day of appearance of post-traumatic headache (PTH) after minor head trauma (MHT) in 25 children aged 3–14 years.

**Figure 4 children-10-00534-f004:**
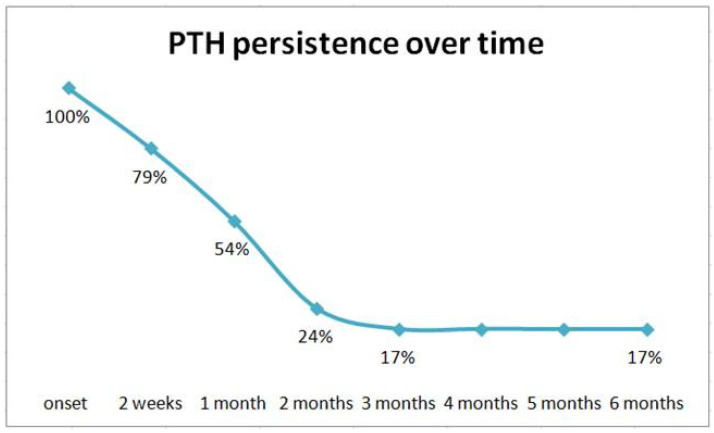
Persistence of post-traumatic headache (PTH) in 24 * children aged 3–14 years followed up for 6 months after minor head trauma. * 1 out of 25 children complaining PTH was lost at follow-up.

**Table 1 children-10-00534-t001:** Characteristics of 224 pediatric patients referred to the pediatric emergency department (PED) for minor head trauma (MHT). Instability or dizziness were considered when persistent regardless of headache attacks. In the case of gait instability, orthopedic injuries were ruled out by physical and/or radiological examination. Amnesia includes loss of memory of the events both preceding and following MHT. * fall from >1.5 m; ** fall from <1.5 m; ° listed further in the same column. CT: computer tomography; SD: standard deviation.

Patients	*n*	224
Age	Mean ± SD	6.7 ± 3.2
	Median	6
Sex	M/F	149/75
HT characteristics, *n* (%)		
mechanism	Car accident	13 (5.8%)
	High fall *	13 (5.8%)
	Low fall **	122 (54.5%)
	Play/sports strike	19 (8.5%)
	Head struck by object	57 (25.5%)
Location	Frontal	95 (42.4%)
	Non-frontal	129 (57.6%)
Symptoms and physical findings at PED, *n* (%)		
	Headache alone	43 (19.2%)
	Headache associated with other symptoms and findings °	25 (11.2%)
	Visual disturbances	8 (3.6%)
	Hematoma	26 (11.6%)
	Swelling	44 (19.6%)
	Soft tissue localized pain in the trauma site	97 (43.3%)
	Altered mental status	18 (8.0%)
	Amnesia	6 (2.7%)
	Nausea and/or vomiting	29 (12.9%)
	Instability or dizziness	13 (5.8%)
Glasgow Coma Scale ≥ 14		100%
Head CT scan in PED, *n* (%)		5 (2.2%)
Time spent in PED, median (SD)		2:26 h (2:28)
Time between MHT and PED consultation, median (SD)		22:57 h (7:27)
Triage system priority, *n* (%)	High risk	23 (10.3%)
	Low risk	201 (89.7%)

**Table 2 children-10-00534-t002:** Characteristics of 25 patients referred to the pediatric emergency department (PED) for minor head trauma (MHT) and who developed post-traumatic headache (PTH). PECARN: Pediatric Emergency Care Applied Research Network. * Some patients presented multiple physical findings; ° 1 patient was lost at follow up; # 1 patient experienced multiple triggers.

Characteristics		N (%)
Risk assessment in MHT according to PECARN score	High risk	3 (12.0%)
	Low risk	22 (88.0%)
Mechanism in low risk MHT	Fall	13 (59.1%)
	Head struck	7 (31.8%)
	Play/sports strike	2 (9.1%)
Location of MHT	Frontal	14 (56.0%)
	Others	11 (44.0%)
Physical findings *	Localized pain	15 (60.0%)
	Altered mental status	1 (4.0%)
	Amnesia	3 (12.0%)
	Nausea and/or vomiting	5 (20.0%)
	Instability or dizziness	2 (8.0%)
PTH episodes occurrence °	>1 per week	15 (62.5%)
	1 per week	8 (33.3%)
	<1 per week	1 (4.2%)
Duration of episodes °	<3 h	16 (66.7%)
	>3 h	3 (12.5%)
	Variable	5 (20.8%)
Time of the day °	Afternoon	9 (37.5%)
	Morning	2 (8.3%)
	Awakening	5 (20.8%)
	Evening	3 (12.5%)
	Variable	5 (20.8%)
Trigger °#	Physical exercise	6 (25.0%)
	Food ingestion	1 (4.2%)
	None	18 (75.0%)
Pain medication °	Ibuprofen or paracetamol	15 (62.5%)
	Not needed	9 (37.5%)

**Table 3 children-10-00534-t003:** Risk factors for post-traumatic headache (PTH) development in patients referred to the pediatric emergency department (PED) for minor head trauma (MHT). Data based on the initial telephone interview of 193 patients. Frequencies relate to the total number of patients with the same characteristics and in the same line (e.g., male patients with no PTH are 117, meaning 90.6% of all males). In the case of gait instability, orthopedic injuries were ruled out by physical and/or radiological examination. Amnesia includes loss of memory of the events both preceding and following MHT. Mechanisms of MHT, high-risk: motor vehicle collision with patient ejection, death of another passenger, or rollover; pedestrian or bicyclist without helmet struck by motorized vehicle; falls >1.5 m; or head struck by high-impact object [3]. School age: ≥6 years old; pre-school age: <6 years old; RR: relative risk; CI: confidence interval; CT: computed tomography; LOC: loss of consciousness. * Yates’ correction has been applied, ^§^ Fisher’s exact test.

Characteristic		No PTH (*n* = 168)	PTH (*n* = 25)	*p* Value	RR	CI- 95%
Sex, patients	Male (%) Female (%)	117 (90.6) 51 (79.7)	12 (9.4) 13 (20.3)	0.055 *		
Age	Schooler Preschooler	89 (81) 79 (95.2)	21 (19) 4 (4.8)	0.0042 ^§^	4	1.4–11.4
	Female Schooler Female Preschooler	21 (61.7) 30 (100)	13 (38.2) 0	<0.001 ^§^		
Location of MHT	Frontal Others	96 (87.3) 72 (86.7)	14 (12.7) 11 (13.2)	0.914 ^§^		
Scalp edema in PED	Present Absent	34 (86.8) 134 (87)	5 (13.1) 20 (13)	0.978 ^§^		
Hematoma	Present Absent	21 (91) 147 (86.5)	2 (9) 23 (13.5)	0.744 ^§^		
Mechanism of MHT	High risk Low risk	23 (88.5) 145 (86,8)	3 (11.5) 22 (13.2)	0.817 ^§^		
Head CT scan	Yes No	4 (66) 164 (87.7)	2 (33) 23 (12.3)	0.131 ^§^		
Any headache in PED	Yes No	28 (70) 140 (91.5)	12 (30) 13 (8.5)	0.0008 *	3.5	1.7–7.1
Headache in PED with other symptoms	Yes No	41 (70) 127 (94)	17 (30) 8 (6)	<0.00001 *	4.9	2.7–10.8
Headache and/or amnesia in PED	Yes No	29 (67.4) 139 (92.6)	14 (32.6) 11 (7.3)	0.000044 *	4.4	2.2–9.1
Headache and/or nausea/vomiting in PED	Yes No	34 (69.4) 134 (93)	15 (30.6) 10 (7)	0.000059 *	4.4	2.1–9.2
Headache and/or gait instability in PED	Yes No	31 (68.8) 137 (92.6)	14 (31.2) 11 (7.4)	0.000101 *	4.19	2–8.6
Headache and/or LOC in PED	Yes No	38 (73) 130 (92.2)	14 (27) 11 (7.8)	0.001082 *	3.5	1.7–7.1
LOC and/or amnesia and/or instability and/or nausea/vomiting in PED	Yes No	23 (79.3) 145 (88.4)	6 (20.7) 19 (11.6)	0.296 ^§^		

**Table 4 children-10-00534-t004:** Patterns of headache and characteristics among 24 patients who developed post-traumatic headache (PTH) and completed 6 month follow-up.

Characteristic		Migraine	Tension-Type
Sex, patients	Male Female	1 5	9 9
Age	Mean ± SD	6.4 ± 3	7.9 ± 2.5
Family history of headache	Yes No	3 3	11 7
Intensity	Mild Moderate Severe	0 6 0	11 7 0
Pain quality	Pulsating Continuous Gravative/burdensome Not defined	3 3 0 0	1 10 6 1
Location	Unilateral Bilateral Global Variable	1 3 2 0	4 6 7 1
Associated or prodromal symptoms *	Photophobia Phonophobia Photo- and Phonophobia Nausea or vomiting None	5 4 0 0 1	3 4 0 1 11

* Some patients experienced multiple prodromal symptoms.

**Table 5 children-10-00534-t005:** The angular coefficients of linear regression lines for post-traumatic headache (PTH) incidence (dependent variable), ODDs ratio, and the risk factors (independent variables, rows) are shown. In the case of gait instability, orthopedic injuries were ruled out by physical and/or radiological examination. Amnesia includes loss of memory of the events both preceding and following MHT. MHT: minor head trauma; PED: pediatric emergency department; CT: computed tomography.

	Coefficients	ODDs Ratio	Sign.
Sex	−1.148	0.317	0.015
Age	0.089	1.093	0.235
Location of MHT	−0.487	0.614	0.343
Scalp edema in PED	−0.180	0.835	0.764
Hematoma	−0.824	0.439	0.346
Mechanism	0.309	0.734	0.683
Head CT scan	0.337	1.400	0.804
Any headache in PED	1.956	7.073	0.001
Loss of consciousness	−1.925	0.146	0.113
Amnesia in PED	1.844	6.320	0.136
Nausea/vomiting in PED	−0.777	0.460	0.274
Gait instability in PED	0.286	1.331	0.774
Constant	−0.370	0.691	0.887

## Data Availability

All clinical data and material are available in our pediatric unit.

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
