# Peer review of "Post-Traumatic Headache in Children after Minor Head Trauma: Incidence, Phenotypes, and Risk Factors"

_children, 2023, doi:10.3390/children10030534_

Round 1

Reviewer 1 Report

Very interesting paper!

Introduction

There are a few more risk factors for prolonged recovery, which may be useful to mention

Methods

During followup phone calls, was the prevalence of repeated head injury assessed or excluded? This is fairly common in my practice, and definitely affects outcomes.

Results

1. Only 5.8% complained of dizziness. This seems very very low compared to my clinical experience. How was the question asked? If patients only had dizziness during headaches, was this counted separately?

Table 3

I find the percentages reported to be a bit confusing- it may be more helpful to report the percentages of PTH vs No PTH, rather than the percentages of male vs female with PTH for example

- What do you mean by # of patients with tumor or edema? (bottom of page 8). Is this scalp edema or edema or tumor on imaging (as it was stated earlier that no patient required followup imaging)?

- Did you differentiate amnesia preceding the injury from amnesia following the injury?

It may be helpful to describe what is considered high-risk vs low-risk mechanisms of injury 

Did you look at types of injury (fall vs motor vehicle accident) as predictors of outcomes?

Table 5

For gait instability, were orthopedic injuries ruled out?

Discussion

You mention return to activity as a factor that may shorten recovery times, but in your conclusion you state that avoiding physical activity may limit the onset of PTH. This doesn't seem supported by your study, and is generally contraindicated in practice as increasing physical activity despite symptoms anecdotally seems to be more helpful (the kids that don't do any physical activity seem to have a harder time if they remain sedentary longer). 

Reviewer 2 Report

Thank you for inviting me to review the manuscript entitled, “Post-traumatic headache in children after minor head trauma: incidence, phenotypes, and risk factors.”  In the study described in this paper, children ages 3-14 who were evaluated for head trauma in an emergency department were surveyed (via phone interview or medical examination) at approximately 15 days, 3-months, and 6-months post-injury to monitor prevalence and characteristics of post-traumatic headache. The main conclusions from the study are that headache at time of presenting with the initial head trauma, with or without nausea and dizziness, is the main risk factor for post-traumatic headache, and that post-traumatic headache is more common in females and in relatively older children.  

The study addresses a relatively understudied topic area in pediatric headache and thus represents a potentially useful contribution to existing literature. The methods are generally well described. Overall, the paper is well-written overall, with adequate justification given for the focus and design of the study. My main concern is that there are recurring errors in the numbers reported; the inconsistency in numbers is worrisome for the validity of the rest of the data reported. Specific comments/questions are provided below:

(1) Although a rationale is provided for excluding children younger than 3, what was the rationale for setting a maximum age of 14 for the study?

(2) Please clarify how “headache persistence” (or lack thereof) is defined for the study (as used in Figure 1). Specifically, how was it determined if the patient would be followed through to the 6-month timepoint – for example, at the 3-month timepoint, would a patient who reported one headache episode since the initial phone interview still be considered as having “headache persistence?”  

(3) The flowchart in Figure 2 shows that of the 224 patients enrolled, 35 were lost to follow-up (which would leave n=189). In section 3.2 (line 137), however, a denominator of n=193 is used to calculate the proportion of patients who developed PTH. This same denominator appears to be what is used in Table 3. Please reconcile and double-check numbers throughout, as differences in denominators would affect many of the other values reported (and would also affect the reported prevalence of post-traumatic headache mentioned in the Discussion).

(4) Relatedly, the numbers reported for “tumor/edema at PED” in Table 3 add up to less than n=193. Conversely, the numbers reported for “LOC and/or Amnesia and/or Instability and/or nausea/vomiting in PED” add up to more than n=193. Please reconcile.

(5) In Table 3, the p-value associated with the first row (gender) is stated to be “p>0.05.” If this is based on a chi-square test, this result seems incorrect and is inconsistent with regression results reported later. Please double-check what is reported. Relatedly, I recommend putting in p-values for ALL results into this table, including non-significant effects, rather than indicating "p>0.05" for non-significant results.

(6) Consider doing a supplemental analysis in which all n=35 lost to follow-up did and did not develop a post-traumatic headache; this would help in defining the potential range of “true” prevalence if the entire enrolled sample was maintained.

(7) Consider mentioning as a limitation that the sample comprised twice as many males as females. Although this may reflect expected variances in the likelihood of minor head trauma, the relatively low sample of females (smaller denominator) relative to males may partly bias the reported inflated risk of post-traumatic headache in females.

(8) Minor point to clarify:  Post-traumatic headache is defined as a secondary headache that starts within 7 days from the injury.  For this study, a headache would still be considered a post-traumatic headache if the minor head trauma exacerbated the primary headache symptoms.  In this case, how is (secondary) post-traumatic headache distinguished from a head injury being a trigger for a primary headache condition? 

(9) Minor point:  Line 85-86:  “Children complaining headache…” should be “Children complaining of headache…”.  Also, the sentence at lines 119-120 is oddly worded:  “Pre-school children were 99, whereas school-age were 129.” 

(10) Minor point:  For Figure 2, in addition to including the sample size of those who developed post-traumatic headache and completed through 6-month follow-up, I suggest also including the sample size for those patients who did not develop post-traumatic headache and who were therefore not followed any further. 

(11) Minor point:  For Table 3, third row, the words “Female Preschooler” gets divided between two rows and may add to confusion about what effect is being reported. If possible, I recommend expanding the column so that “Female Preschooler” fits on the same row; alternatively, remove the capital letter “P” from preschooler (“Female preschooler”).

(12) Minor point: For Table 3, consider adding to the table title that these data were based on the initial telephone interview (this will help clarify why n=25 for post-traumatic headache here, whereas n=24 for post-traumatic headache in Table 4).

Reviewer 3 Report

Dear Authors,

congratulations on your research Post-traumatic headache in children after minor head trauma:  incidence, phenotypes, and risk factors.

However, in my opinion, this manuscript needs a minor revision, according to the following remarks:

Introduction: it is too  concise. Please elaborate more about MHT.

M&M: Lines 58-115 should be divided into the following pharagraphes: study design, recrutiment to the study, inclusion and exclusion criteria, ethical issues, statistical analysis.

Discussion: Could the authors elaborate more about future research implications. Could the authors elaborate more about practical clinical implications?

Round 2

Reviewer 2 Report

Overall, my comments from the review of the initial version of the manuscript have been adequately addressed.  I had two remaining suggestions that are considered minor:

(1) The single sentence paragraph on lines 54-56 now seems to be out of place and should be better integrated with the rest of the introduction (e.g., integrate within the first paragraph of the Intro)

(2) I appreciate the effort to respond to a prior comment about indicating “true incidence of PTH” if assuming that all those lost to follow-up did actually develop PTH versus assuming all those lost to follow-up did not develop PTH.  However, the wording of this addition is somewhat awkward (lines 199-201).  Consider instead something like, “If assuming that all patients lost to follow-up developed PTH, the incidence of PTH would increase to 26.3% (60/228); if instead assuming that all patients who were lost to follow-up did not develop PTH, the incidence of PTH would decrease to 10.9% (25/228).”

Author Response

“Overall, my comments from the review of the initial version of the manuscript have been adequately addressed.  I had two remaining suggestions that are considered minor:

  • The single sentence paragraph on lines 54-56 now seems to be out of place and should be better integrated with the rest of the introduction (e.g., integrate within the first paragraph of the Intro)”

Reply: this sentence was integrated with the first paragraph of the introduction.

“(2) I appreciate the effort to respond to a prior comment about indicating “true incidence of PTH” if assuming that all those lost to follow-up did actually develop PTH versus assuming all those lost to follow-up did not develop PTH.  However, the wording of this addition is somewhat awkward (lines 199-201).  Consider instead something like, “If assuming that all patients lost to follow-up developed PTH, the incidence of PTH would increase to 26.3% (60/228); if instead assuming that all patients who were lost to follow-up did not develop PTH, the incidence of PTH would decrease to 10.9% (25/228).””

Reply: thank you for improving our sentence. It was changed according to your suggestion.